# Chemical Composition, Functional and Anticancer Properties of Carrot

**DOI:** 10.3390/molecules28207161

**Published:** 2023-10-19

**Authors:** Luigi Mandrich, Antonia Valeria Esposito, Silvio Costa, Emilia Caputo

**Affiliations:** 1Research Institute on Terrestrial Ecosystems-IRET-CNR, Via Pietro Castellino 111, 80131 Naples, Italy; luigi.mandrich@cnr.it; 2Institute of Genetics and Biophysics-IGB-CNR, “A. Buzzati-Traverso”, Via Pietro Castellino 111, 80131 Naples, Italy; antonia.esposito@igb.cnr.it (A.V.E.); costasilvio7@gmail.com (S.C.)

**Keywords:** *Daucus carota* L. phytochemicals, anti-proliferative compounds, anti-metastatic compounds, polyunsaturated fatty acids (PUFAs)

## Abstract

Plants are a valuable source of drugs for cancer treatment. *Daucus carota* has been investigated for its health properties. In particular, *Daucus carota* L. subsp. *Sativus*, the common edible carrot root, has been found to be rich in bioactive compounds such as carotenoids and dietary fiber and contains many other functional components with significant health-promoting features, while *Daucus carota* L. subsp. *Carrot* (Apiacae), also known as wild carrot, has been usually used for gastric ulcer therapy, diabetes, and muscle pain in Lebanon. Here, we review the chemical composition of *Daucus carota* L. and the functional properties of both edible and wild carrot subspecies. Then, we focus on compounds with anticancer characteristics identified in both *Daucus carota* subspecies, and we discuss their potential use in the development of novel anticancer therapeutic strategies.

## 1. Introduction

Carrot (*Daucus carota* L.) is a biennial herbaceous species belonging to the Apiaceae family [1]. It is one of the most popular root vegetables grown worldwide.

The carrot is composed of the umbel, the stem, and the root (Figure 1A).

The stem under the white flower umbrella can reach a height of about 1 m (Figure 1B). The roots represent the most commonly eaten part of the carrot; they are greatly enlarged and sweet with good storage ability. The root is formed by the peel or periderm, the pulpy outer cortex or phloem, and the inner core or xylem, as illustrated in Figure 1A.

Carrots, based on the pigmentation of their roots, have been classified into eastern and western carrots [2]. Most eastern carrots have purple roots, while only some have roots that are yellow in color. On the other side, western carrots have orange, red or white roots.

Among *D. carota* L. subspecies, the best known and studied are *D. carota* subsp. *Sativus* (or domestic carrot) and *D. carota* subsp. *carota* (or wild carrot) (Figure 1C). These subspecies differ from each other mainly in the color and thickness of the root, as well as in their taste. Domestic carrot roots exhibit a wide range of colors, from white to purple, as illustrated in Figure 1D–G. They are thick and have a sweet taste. Wild carrots, also known as Queen Anne’s lace, have thin, white roots with a bitter taste.

Based on a popular myth, domestic carrots seem to be derived directly from the wild ones. Further, their similar features, such as the leaf pattern, the odor, and the way that they grow, support this myth [3]. Moreover, wild and domestic carrots intercross freely, and they are predominantly cross-pollinated by a large diversity of insects, so the gene-flow frequency may be very high when both the spatial distribution and flowering overlap.

Recently, a review reported several studies performed on the biology and origins of carrots, their cultivation, their chemical composition, and “omic” analyses [4], supporting their abundant contents, including beneficial nutrients for human health. For instance, as with other fruits and vegetables, the consumption of carrots is recommended by global dietary guidelines [5] advocating five portions of fruits and vegetables per day. 

Here, we focus on the chemical composition of carrots and on the main bioactive compounds, highlighting their health-promoting functions and their anticancer properties. Potential further work in carrot research is also discussed.

### Daucus carota L. Chemical Composition

The chemical constituents of carrots have been investigated, and their contents have been measured. Different values for each of them have been determined, depending on the different carrot varieties examined [6], as summarized in Table 1.

The moisture content has been determined to range in value from 86 to 89%. Minerals like calcium (Ca), phosphorus (P), potassium (K), magnesium (Mg), manganese (Mn), iron (Fe), and sodium (Na) have been detected in carrots. Traces of copper (Cu, 0.02 mg/100 g) and zinc (Zn, 0.2 mg/100 g) have also been reported. Among them, it has been observed that the iron, sodium, and magnesium quantities are different depending on the carrot variety. Moreover, the mineral content value represents one of the important parameters used to define the quality of this vegetable [7,10,11]. Carbohydrates, such as simple sugars like fructose, glucose, and sucrose and small amounts of starch and fiber, have also been found in carrots [8,12]. Their contents may change in different carrot varieties and are affected by environmental and storage conditions [13]. Thus, the carrot, as with other vegetables, is considered a source of prebiotics by CODEX Alimentarius [8]. Dietary fiber content is also very high in carrots. Such fiber plays an important role in human health, promoting healthy bowel function and decreasing cholesterol levels, as well as the risk of heart diseases. The crude fiber found in carrot roots is mainly from cellulose, while hemicellulose and lignin are detected in lower quantities [7]. Moreover, the cellulose content has been found to vary from 35 to 48%, depending on the carrot variety. Very small amounts of succinic acid, α-ketoglutaric acid, lactic acid, and glycolic acid have also been detected. The average nitrate and nitrite contents in fresh carrots have been reported to be about 40 and 0.41 mg/100 g, respectively [14], while appreciable amounts of thiamin, riboflavin, niacin, folic acid, and vitamin C have been found [14]. Further, carrot roots are a good source of anthocyanins: their contents may vary from trace amounts in pink cultivars to 1750 mg/kg in black carrots; in particular, cyanidin 3-(2-xylosylgalactoside), cyanidin 3-xylosylglucosylgalactoside, and cyanidin 3-ferulylxyloglucosyl galactoside are the major ones identified. On the other hand, the glutamic acid content and the buffering action of free amino acids are mainly responsible for the carrot taste.

In addition, carrot roots are a good source of vitamins, the contents of which represent another important parameter, together with the mineral content, used to define the quality of this vegetable [10]. The main vitamin types identified in carrot varieties are vitamin A (derived from β-carotene), which plays an important role against night blindness and is able to increase immune system functions [15], and vitamin E (191–703 μg/100 g), derived from α-tocopherol and critical for cell signaling, gene expression, and cell membrane stability in the human body. Moreover, carrot varieties are rich in vitamin B derivatives (thiamine, riboflavin, cobalamin, and pyridoxine) [16], which are important for cell growth and brain and digestion system functions.

On the other hand, carrots are rich in organic acids. The most well known among them is ascorbic acid (also known as vitamin C). It exhibits high antioxidant activities responsible for its roles in controlling blood pressure, preventing iron deficiency, and boosting immune system functions [8]. However, many others support healthy body functions; for example, benzoic acid and hydroxycinnamic acid show antibacterial and anti-inflammatory activity, respectively, while gallic acid acts as an anti-mutagenic factor. Moreover, acetic, succinic, citric, lactic, and malic acids and their salts stimulate iron absorption [8].

Furthermore, carrots are rich in C_17_-polyacetylenes (PAs), a group of oxylipins derived from crepenynic and dehydrocrepenynic acids, also known as “unusual” polyunsaturated fatty acids. The most abundant PAs detected in carrots are falcarinol, falcarindol, and falcarindol-3-acetate. The total PA contents and their relative distribution are variable, depending on the source (cultivated orange carrot or wild *D. carota* ssp.), the storage time, root size, age, and physiological stage [9].

## 2. *Daucus carota* L. Bioactive Components with Health-Promoting Function or Phytochemicals

Several epidemiological studies have reported that the increased consumption of carrots is related to a decreased risk of different diseases, such as chronic diseases, cardiovascular diseases, age-related macular degeneration, and cancer [17,18]. These health benefits of carrots are mainly due to a wide range of bioactive compounds among their constituents, also called phytochemicals, including carotenoids, phenolics, flavonoids, vitamins, anthocyanins, and fatty acids [19].

### 2.1. Carotenoids

Carotenoids are compounds with low solubility in water. They belong to the tetraterpene family (C40-based isoprenoid), responsible for the yellow, orange, or red color of fruits, leaves, and flowers. Most carotenoids have a central carbon chain with alternating single and double bonds and carry various cyclic or acyclic end groups [20,21]. Generally, in carrots, their contents range from 16 to 38 mg/100 g [22].

To date, about 700 carotenoids derived from vegetables and fruits have been described and have been grouped into carotenes and xanthophylls, carrying their specific color range from yellow to red [23]. Moreover, carotenoids can be oxidatively cleaved by dioxygenases, and the resulting carotenoids are classified as apocarotenoids [23]. In carrot roots, the major carotenoids are carotenes, particularly β-carotene (75%), α-carotene (23%), and lutein (1.9%), and xanthophylls, namely, β-cryptoxanthin, lycopene, and zeaxanthin. Beta-carotene content represents about 80% of total carotenoids contained in domestic carrot roots (Figure 2).

Among all identified carotenoids, approximately 40 are present in a typical human diet, and of these, only 20, including β-carotene, α-carotene, lycopene, lutein, and cryptoxanthin, have been identified in human blood and tissue [20]. Carotenoids are accumulated mainly in the root. However, their contents are different in different root tissues of carrots with different root colors [24]. These differences seem to be related to the different expression patterns of carotenoid biosynthesis genes in carrot-specific tissues. It was observed that in the roots of orange and purple carrots, the content of carotenoids in the phloem is higher than that observed in the xylem, while in the red ones, the contents of carotenoids in the phloem and xylem are similar [4]. However, gene expression only partly describes the differences in carotenoids’ accumulation in the various carrot tissues [25]. In general, orange carrots mainly contain α- and β-carotene; yellow ones contain lutein, while red ones contain lycopene [26]. In particular, the carotenoid amount has been estimated in a range from 469 to 605 μg/100 g in yellow and purple carrots, while it is 10 times higher in orange carrots. Whereas 170 mg/kg of β-carotene was found in dark orange carrots, only 3.2 mg/kg is estimated in purple carrots [27].

Moreover, it has been demonstrated that the content of carotenoids is not only affected by different genotypes but also dependent on environmental factors, as well as storage conditions and temperature [28]. It has been reported that, although β-carotene is present in carrots at the highest amount compared to other fruits and vegetables, it is lost during processing and storage. Recent studies concerning its retention in carrots revealed that reducing the water activity of β-carotene results in poor shelf-life compared to canning or freeze-drying processes. During both of these processing methods, the *trans* form of β-carotene in carrots is replaced by the *cis* form, resulting in more effective β-carotene retention during storage. Furthermore, it was observed that β-carotene in carrot juice extract has a half-life of only 2 days at room temperature, which can be increased up to 6 months by encapsulation methods.

Although controversial findings have been reported, epidemiological studies have provided useful data on the possible protective roles of foods or food components in disease prevention [29,30]. Several studies showed an association between the high dietary intake of foods rich in carotenoids (particularly β-carotene and lycopene) and a decreased risk of developing cancer, particularly forms of lung and stomach cancer [29,30,31], as well as a decreased risk of developing cardiovascular problems [30]. Furthermore, among carotenoids, it has been suggested that lutein and zeaxanthin also have a protective role against the development of some eye diseases [32]. Carotenoids have also been reported to be potential inhibitors of Alzheimer’s disease [33].

### 2.2. Phenolics

Carrots also contain phenolic acids (*p*-hydroxybenzoic, caffeic, and chlorogenic) and flavonoids (anthocyanins). Both phenolic acids and flavonoids, along with tannins, are phenolic compounds (PCs) and are classified according to the number of attached phenolic hydroxyl groups and the structural elements linking benzene rings [34], as shown in Figure 3.

PCs can be either free, conjugated (to sugars and low-molecular-weight compounds), or insoluble (BP). Insoluble PCs are covalently linked to the structural components of the cell wall [35].

In carrots, phenolic compounds have been found in high concentrations in the root periderm tissues. However, their concentrations decrease from the peel (periderm) to the xylem of carrot root tissues. Although the peel represents only 11% of the total fresh weight of the carrot, it contains a higher concentration of total PCs (54.1%) compared to the phloem (39.5%) and xylem (6.4%). In particular, chlorogenic acid, the major hydroxycinnamic acid derivative, is the most abundant among the total PCs identified in various carrot tissues, accounting for 42.2% to 61.8% of them [34].

Different PC contents have been found among the orange-, purple-, yellow-, and white-colored carrot varieties [12]. In particular, the total PC amount was higher in purple carrots (74.6 mg/100 g) compared to the corresponding values observed in orange, yellow, and white varieties, in which it ranged from 7.72 to 16.2 mg/100 g [12]. Moreover, it has been reported that PC levels change when exposed to stress. For instance, the PC content increased during wounding, radiation, and storage, likely as a consequence of phenylalanine ammonia lyase (PAL) activation, involving ATP and reactive oxygen species as signaling molecules. Great differences in phenolic content were also related to atmospheric changes [36]. Patterns of PC accumulation varied across carrots depending on the different geographic areas where they were grown. Furthermore, the quantity of phenolic compounds increases very slowly when shredded carrots are stored in polypropylene film bags or in controlled atmospheres containing 30% CO_2_ and/or 0% O_2_. An increase in trans-5′-caffeoylquinic acid has been reported in shredded carrot tissue stored at 4 °C after mechanical damage caused by the cooling process. On the other hand, chlorogenic acid accumulation has previously been observed in carrots following an infection or as a common response to various stressors. Furthermore, the accumulation of PCs in shredded carrots stored in the air appears to be associated with an increase in PAL activity [37]. 

It is now known that reactive oxygen species (ROS) play an important role in the biosynthesis of phenolic antioxidants in wounded carrots. It was observed that, in wounded carrots, the higher the intensity of the wound and the carrot storage temperature, the higher the production of ROS and the higher the accumulation of phenols. Furthermore, treatment with a ROS inhibitor, in addition to inhibiting ROS production, suppresses the activities of key enzymes in the phenylpropanoid pathway (phenylalanine ammonia lyase, PAL; cinnamate-4-hydroxylase, C4H; coenzyme 4-coumarate A ligase, 4CL), leading to limited accumulation of phenols in shredded carrots. Conversely, ROS elicitor treatment promotes ROS generation, enhances PAL, C4H, and 4CL activities, and induces phenol accumulation. Thus, ROS play a critical role in mediating wound-induced phenol accumulation in carrots. These data indicate that high temperature increases the accumulation of phenols by inducing the generation of ROS [36]. It has been demonstrated that PCs are involved in imparting color, especially anthocyanins, as well as in the repair of wound damage and in defense against microbial invasion [38]. Further, these compounds have been observed to have sensory roles and to act as food preservatives. PCs have also been reported as responsible for the antioxidant properties of carrots. They are able to donate a hydrogen atom or an electron to a free radical to form stable intermediate products, scavenging free radicals, decomposing oxidation products, and chelating metal ions [39].

It has been shown that carrots exhibit different antioxidant properties depending on their different colors. Several studies consistently reported that among different carrot colors, purple ones exhibited the highest antioxidant activity due to their higher phenolic compound concentration [18].

Furthermore, phenolic compounds such as anthocyanins and flavonoids have been observed to have good antioxidant power compared to standard phenols. These compounds, along with their role in scavenging or inactivation, are able to interfere with the production of free radicals [40]. Interestingly, the carrot phenolic extracts from peels exhibited much higher antioxidant and diphenyl-1-picrylhydrazyl (DPPH)-scavenging activities than the extracts from the phloem and xylem.

### 2.3. Vitamins

Vitamins are also considered phytochemical compounds. These compounds have heterogeneous structures and are typically classified as either water-soluble or fat-soluble [41], as shown in Figure 4.

The most abundant vitamins detected in edible carrot roots are water-soluble vitamin C (L-ascorbic acid) and fat-soluble vitamin E isomers: α-tocopherol, γ-tocopherol, α-tocotrienol, and γ-tocotrienol.

α-Tocopherol is the major vitamin E isomer and is found in all carrot genotypes. α-Tocopherol has been found in large amounts in the outer tissues compared to the inner tissues of all genotypes. Generally, its content is higher in genotypes with purple roots than those with other root colors. Further, this compound is not very stable, and its content significantly decreases in fresh carrots. α-Tocotrienol is the second one detected in all carrot genotypes, except in carrots with white roots. Less abundant is γ-tocopherol. It has been found only in white carrot genotypes, and its content is much lower than those of α-tocopherol and γ-tocotrienol. Finally, γ-tocotrienol, the least dominant vitamin E isomer, was instead observed in two genotypes of white carrots. The total vitamin E content is higher in the outer tissues compared to the inner tissues of almost all genotypes. However, the purple carrot genotype shows the highest total vitamin E content in the inner and outer tissues compared to all of the other carrot genotypes [42]. 

Vitamin C is a heat-labile vitamin: in dehydrated products, it is lost due to its degradation into di-ketogulonic acid. Thus, vitamin C, due to its low stability, is used as an indicator of the effects of carrot processing on the degradation of its nutrients. It has been observed that in the fresh product, vitamin C begins to rapidly degrade, and refrigeration slows down this process. Furthermore, if the product undergoes blanching before freezing, it loses a significant amount of vitamin C, which is instead reduced after steam blanching [43]. However, the stability of vitamin C in carrots has been shown to be influenced by various factors during the pre-harvest, harvest, and post-harvest handling stages. One of the strategies used to maintain the stability of vitamin C is to prevent the oxidation of L-ascorbic acid (L-AA) to dehydro-L-ascorbic acid (DHAA), which is catalyzed by oxidative enzymes such as ascorbic acid oxidase (AAO) [44].

Vitamin E has been observed to play a protective role in various diseases, such as atherosclerosis, cancer, cataracts, Alzheimer’s disease, and cardiovascular diseases [45]. It acts as a membrane stabilizer and a lipid-soluble antioxidant, as well as favors membrane repair by preventing the formation of oxidized phospholipids; it protects against photo-aging and skin cancer [46]. Further, α-tocopherol is the main antioxidant of the human epidermis and, for this reason, represents an early and sensitive indicator of environmental oxidative damage [47].

Furthermore, vitamin C, a water-soluble antioxidant, acts as a cofactor in numerous enzymatic reactions in the body based on its ability to donate two electrons. Vitamin C is essential for maintaining proper skin structure and function, participating in the formation of collagen cross-links during the hydroxylation of proline and lysine. It also plays a critical role in maintaining adequate skin hydration, and it enhances the synthesis of ceramide-lipid compounds in the stratum corneum [47]. Further, vitamin C inhibits melanogenesis and the enzyme tyrosinase [48]. Ascorbic acid is oxidized, transforming into the ascorbate anion, which can continue with electron donation, leading to its conversion to the ascorbate free radical, which shows greater stability compared to other free radicals. It is then converted to dehydroascorbic acid [47,49].

### 2.4. Fatty Acids and Their Derivatives

Fatty acids are gaining more attention for their role as phytochemicals. They are either saturated or unsaturated carboxylic acids with carbon chains varying between 2 and 36 carbon atoms. Polyunsaturated fatty acids (PUFAs) are characterized by a pentadiene configuration of double bonds (Figure 5).

The synthesis of fatty acids occurs in the cytoplasm, starting from two-carbon precursors, with the aid of an acyl transporter protein, NADPH, and acetyl-CoA-carboxylase, while their degradation occurs in the mitochondria and is accompanied by energy release [19]. 

In carrots, unsaturated fatty acids make up about 70% of the total fat content compared to saturated ones. Palmitic acid, which is a saturated fatty acid, and linoleic and linolenic acids, which are unsaturated fatty acids, represent the main fatty acids present in carrot seeds. Moreover, the two unsaturated fatty acids constitute the precursors of aliphatic C_17_ acetylenic oxylipins, the main polyacetylenes isolated from carrots [50], and, among these fatty acid derivatives, the falcarinol-type oxylipins are the most common ones identified in carrots, as illustrated in Table 1 and Figure 5. 

The high content of polyunsaturated fatty acids (PUFAs) in carrot roots has been associated with their health potential. It has been reported that PUFAs exhibit antioxidant, anti-inflammatory, and antipyretic effects [42,51]. Interestingly, it has been reported that, among PUFAs, falcarinol-type oxylipins exhibit cytotoxic and anticancer activities [9,50], as described in more detail below.

## 3. Carrot Compounds and Their Role in Cancer

Several studies have shown that some carrot metabolites are able to induce powerful cytotoxic effects specifically in cancer cells by interfering with important cellular pathways. It has been demonstrated that carrot metabolites can modulate different proteins involved in cell proliferation, apoptosis, epithelial-to-mesenchymal transition, and inflammation, all critical processes responsible for cancer progression and metastasis.

### 3.1. Carrot Compounds and Their Role in Cell Survival, Proliferation, Apoptosis, and Inflammation

It has been shown that some carrot compounds are able to inhibit the cell growth of different cancer cell lines by interfering in cell proliferation, apoptosis, and inflammation processes, as shown in Figure 6.

Among these compounds, acetylenic oxylipins, such as falcarinol (FaOH) and falcarindiol (FaDOH), are upregulated in response to fungal diseases, acting as natural pesticides in carrots [9], and exhibit a diverse range of biological activities in mammals.

It was observed that falcarinol, at low concentrations between 0.004 and 0.4 µM, stimulated the differentiation of primary mammalian cells, while at concentrations above >4 µM, it showed cytotoxic activity [52]. This biphasic behavior, known as hormesis, is typical of bioactive/toxic compounds [53,54], and it has been demonstrated not only for falcarinol but also for falcarindiol and falcarindiol 3-acetate in different normal and tumor cell types [55,56,57,58]. It was observed that the exposure of human colorectal adenocarcinoma cells (Caco-2) to a low concentration of falcarinol (between 0.5 and 10 µM) induced cell proliferation, while exposure to high concentrations (20 µM) inhibited Caco-2 cell proliferation, increasing the number of cells expressing active caspase-3. Moreover, it has been observed that extracts from carrots containing the highest concentration of falcarinol exhibited the highest inhibitory effect on Caco-2 cell growth, supporting the higher cytotoxic potency of falcarinol compared to falcarindiol [57].

Furthermore, the cytotoxicity of falcarinol in Caco-2 cells was found to be synergistically enhanced when used in combination with falcarindiol in a ratio of 1:1, 1:5, or 1:10, suggesting that these compounds may have slightly different mechanisms of action for their cytotoxic effects. In addition, the oxidation of the hydroxyl group at C-3 in falcarinol, leading to falcarinone (4), significantly reduced the cytotoxic effect on Caco-2 cells compared to falcarinol [57].

Although, among the oxylipins, FaOH was found to be the most cytotoxic compound towards leukemic cells, with IC50 values of 12–35 µM, it has been demonstrated that other oxylipins, such as falcarindiol and falcarindiol 3-acetate, isolated from carrots also induced apoptosis in the CCRF-CEM, Jurkat, and MOLT-3 leukemia cell lines at concentrations of 18–68 µM and 23–38 µM, respectively, while falcarindiol-carinol induced apoptosis only in the CCRF-CEM cell line when used at a concentration of 45 µM [58].

In addition, due to their alkylating properties, FaOH and FaDOH are able to induce the cell cycle arrest and apoptosis of tumor cells through their covalent binding to proteins/factors involved in these processes. Interestingly, FaDOH can also induce ER stress in breast cancer cells (MDA-MB-231, MDA-MB-468, and SKBR3) and, subsequently, caspase-dependent cell death. Furthermore, this compound worked synergistically with the approved anticancer drugs 5-fluorouracil and bortezomib in killing breast cancer cells [59]. 

FaOH and FaDOH also exhibit anti-inflammatory activity. It has been observed that, while FaOH exhibited pronounced cytotoxic activity, FaDOH showed enhanced anti-inflammatory activity [60]. They are able to inhibit the transcription factors NF-κB (nuclear factor kappa-light-chain-enhancer of activated B cells) and STAT3 (signal transducers and activators of transcription 3), involved in two major inflammation pathways. It is known that about 25% of all human cancers are linked to chronic inflammation [61]. Helicobacter pylori infections are associated with an increased risk of gastric cancer; there are also associations between human papillomavirus and cervical cancer, hepatitis B or C infection and hepatocellular carcinoma, and inflammatory bowel disease and colorectal cancer (CRC) [62,63,64]. It has been demonstrated that acetylenic oxylipins are able to inhibit NF-κB and the formation of proinflammatory cytokines and inflammatory enzymes such as interleukins (ILs), cyclooxygenases (COXs), and lipoxygenases (LOXs), suggesting that the direct inhibition of these inflammatory mediators may be the mechanism of action adopted by these compounds for the prevention and treatment of cancer (Figure 6). 

Studies in a rat model of CRC demonstrated that (3R)-falcarinol and (3R,8S)-falcarindiol, isolated from carrots selectively, inhibited the expression of COX-2 in tumor tissue as well as TNF-α and IL-6, thus explaining the CRC-preventive effect of carrots [65,66]. Interestingly, there have been reports on a correlation between the contents of both compounds, FaOH and FaDOH, in carrots [67] and the preventive effect of carrot intake on CRC development in a Danish population of 57,053 individuals with a long follow-up period [68,69,70].

Furthermore, the lipophilic nature of these compounds suggests their ability to be rapidly taken up by tumor cells, an important prerequisite for their cytotoxicity and anti-inflammatory activity. However, few in vivo studies have been performed in this regard. A pharmacokinetic study demonstrated that (3S,8S)-falcarindiol (10) and oplopandiol (18) extracted from Oplopanax elatus Nakai, when administered in rats, were rapidly absorbed in vivo [71]. This was also confirmed in a human study, where it was observed that (3R)-falcarinol and (3R,8S)-falcarindiol were rapidly absorbed in humans after the oral administration of carrot juice containing these polyacetylenes [72,73,74].

Moreover, polyacetylenes constitute very promising chemopreventive candidates, as they show low toxicity and no serious side effects known to date. Falcarinol has so far been shown to be the only one that, when administered at high doses in rodents (100 mg/kg by injection), causes neurotoxic effects [75], while falcarindiol does not appear to have any toxic effects [76]. Three preclinical studies on falcarinol and falcarindiol isolated from carrots were performed in vivo by using a rat model of CRC, where rats were induced, by using azoxymethane (AOM), to develop aberrant crypt foci (ACF) and precancerous lesions of CRC [65,66,77,78,79,80]. All three clinical studies demonstrated that falcarinol and falcarindiol inhibited early neoplastic formation in a dose-dependent manner. Adenomas were, in general, smaller in rats treated with falcarinol and falcarindiol, especially at higher doses, compared to the control group [65]. These data clearly demonstrated a dose-dependent chemopreventive effect of falcarinol and falcarindiol on the formation of neoplastic lesions in the colon of rats. The dose for an optimal antineoplastic effect was found to range from 7 to 35 µg of falcarinol and falcarindiol/g feed in a rat model of CRC.

Based on this study, it appears that a preventive dose of FaOH and FaDOH in humans can be achieved with a daily intake of more than 30 g of raw carrots [65,67], which is in line with the data from the above cohort study [67,80]. Of interest, gene expression studies performed by real-time quantitative PCR on selected tumor biomarkers in neoplastic tissue from a rat CRC model revealed that falcarinol and falcarindiol downregulated NF-κB and its downstream proinflammatory markers TNF-α, IL-6, and COX-2, while COX-1 and IL-1 were not significantly affected [65]. 

The anti-inflammatory activity observed in the rat model of CRC may be linked to oxylipin’s ability to activate the Keap1-Nrf2 pathway. It has been recently observed that pretreatment with 5 mg falcarinol/kg twice daily for one week in CB57BL/6 mice with LPS-induced acute bowel syndrome and systemic inflammation reduced the intestinal expression of IL-6, TNFα, INF, STAT3, and IL-10 proinflammatory genes [81]. Additionally, pretreatment with falcarinol induced the upregulation of the cytoprotective enzyme HO-1 in both the intestine and liver, and it decreased basal lipid peroxidation in the mesentery, supporting the cancer-preventive effects of falcarinol and falcarindiol observed in the rat CRC model.

The in vivo anticancer effects of falcarindiol were also evaluated in a xenograft tumor model generated by inoculating the human CRC cell line HCT-116 into athymic nude mice [82,83,84]. After daily intraperitoneal administration of falcarindiol (15 mg/kg), significant inhibition of xenograft tumor growth was observed as early as week 2, with more significant inhibition at weeks 3 and 4 (*p* < 0.01), compared to the tumor growth observed in mice administered only the vehicle (control) [82,84].

In addition, the cytotoxicity of falcarindiol was evaluated on the normal rat small intestine epithelial cell line (IEC-6). It showed no cytotoxic effects on these normal cells, even at a concentration of 20 µM, whereas, at 10 µM, HCT-116 cell growth was inhibited [82], suggesting a specific cytotoxic effect on cancer cells. 

In another study, it was observed that falcarinol, when orally administered in lung cancer mice models, was able to suppress lung cancer growth without detectable toxic side effects. It was demonstrated that this compound was able to inhibit heat shock protein 90 (Hsp90) in non-cancer stem-like and cancer stem-like cells of non-small-cell lung tumors by inducing apoptosis [85]. Hsp90 is known to be overexpressed in many human tumors, and its overexpression is associated with a poor prognosis. In vitro and molecular docking studies showed that falcarinol binds to the N-terminal and C-terminal ATP-binding pockets of Hsp90, resulting in its destabilization and inhibition [85], providing new insight into the possible mechanisms of action for the anticancer properties of falcarinol and the related acetylenic oxylipins. 

Further, more lipid compounds extracted from carrots have been reported to exhibit an anti-proliferative effect specifically on cancer cells. Semi-purified extracts from carrots containing glycerophosphocholine derivatives, GPC (16:0) isomer 2 (3.6%), hydroxyocta-decadienoic acid (92.9%), and monogalactosyl-monoacylglycerol (MGMG) (18:3) (3.4%) compounds showed an anti-proliferation effect only on breast cancer cell lines and not on breast epithelial cell lines [86]. 

Carotenoids have also been reported to inhibit the proliferation of cancer cells. α-Carotene (AC) has been observed to inhibit the proliferation of various human prostate cancer cell lines, such as PC-3, DU145, and LNCaP, as well as that of human neuroblastoma GOTO cells [87,88]. 

Similarly, β-carotene induced a reduction in the proliferation of prostate cancer cell lines in vitro when used at high doses (30 μM). However, the β-carotene effect on these cancer cells was different when used at lower concentrations (3 and 10 μM). For instance, at low concentrations, β-carotene behaved as a pro-proliferation factor in PC3 and LNCaP cell lines. However, this behavior was more pronounced in LNCaP than in PC-3 cells, likely due to the more efficient β-carotene uptake from the medium by androgen-sensitive LNCaP cells [89].

Moreover, an anti-proliferative effect of β-carotene on the MCF7 breast cancer cell line has been reported. In this case, the effect was dose-dependent and induced MCF7 apoptosis through an increase in caspase-3 activity. In particular, β-carotene decreased the expression of the Bcl-2 and PARP anti-apoptotic proteins and that of the NF-kB survival protein. It also inhibited the activation of Akt, resulting in decreased Bad phosphorylation, and of ERK1/2. In addition, β-carotene was demonstrated to be involved in the downregulation, at the protein level, of the antioxidant enzyme superoxide dismutase 2, SOD-2, of its transactivation factor (Nrf-2), and of XBP-1, the endoplasmic reticulum (ER) stress marker, further supporting the anticancer activity of β-carotene in the MCF-7 breast cancer cell line [90].

Interestingly, a specific effect on a triple-negative breast cancer cell line, MDA-MB-231, compared to normal epithelial cells (MCF-12A) was observed when these cells were treated with β-carotene (10 μM). While no effect was observed on MCF-12A, MDA-MB-231 cells showed a slightly modified morphology, from spindle-shaped to an irregular shape, upon β-carotene treatment; they also exhibited reduced proliferation due to β-carotene interference in the JNK pathway without affecting the other examined intracellular signaling pathways (PI3K, ERK/MEK1/MEK2, p38 MAPK) [91]. 

On the contrary, β-carotene did not exhibit anti-proliferative effects on melanoma cells. It was demonstrated that β-carotene alone (1 μM) did not affect BRAF or ERK expression in melanoma cell lines carrying the B-RAF mutation (A2058 cell line), while in combination with B-RAFi (PLX), it enhanced the suppression of B-RAF activation and of the downstream Erk1/2 effector, induced by PLX [92], suggesting its potential use as an adjuvant for target-based therapy. 

Among carrot metabolites, anthocyanins have also been reported to exhibit cytotoxic activities. These compounds are found in high amounts mainly in black carrots (*Daucus carota* ssp. sativus) originating from Turkey and the Middle and Far East. It has been shown that anthocyanins are cytotoxic to the MCF-7, SK-BR-3, MDA-MB-231, HT-29, PC-3, and Neuro-2A cancer cell lines and the normal VERO cell line [93]. It was observed that the highest cytotoxic activity of carrot calli and natural extracts was achieved against the Neuro-2A cell line, exhibiting viability of 38–46% at 6.25 μg/mL concentration, while a significant, high IC50 value (170.13 μg/mL) was measured in the normal VERO cell line, suggesting anthocyanins as ideal candidates for brain cancer treatment. Interestingly, among phenolic compounds, it has been observed that 6-methoxymellein exhibited anti-inflammatory and anticancer activities through NF-κB/IL-6 and IL-8 regulation [94].

Interestingly, recently, it has been found that, by using structure-based virtual screening, anilinonaphthalene from carrots may be used as a potential drug for developing new and effective therapies against multiple cancers due to its ability to selectively bind and inhibit the proto-oncogene c-Kit, a tyrosine-protein kinase involved in the differentiation, proliferation, migration, and survival of cancer cells [95].

### 3.2. Carrot Compounds and Their Anti-Metastatic Role 

Roles for various carrot compounds as anti-metastatic agents have also been reported. For instance, the anti-metastatic role of the alpha-carotene (AC) compound, identified by using 2D cell systems, was reported [96]. The treatment of the human hepatocarcinoma SK- Hep-1 cell line with AC (0.5–2.5 μM) for 48 h induced the significant inhibition of the invasion, migration, and adhesion of these cells in a concentration-dependent manner. Moreover, these AC-induced effects were stronger than those induced upon treatment with β-carotene (BC) in the same cell line at the same concentration (2.5 μM). Alpha-carotene was found to significantly reduce the activities of urokinase plasminogen activator and matrix metalloproteinases (MMP)-2 and -9 while increasing the protein expression of urokinase activator inhibitor, plasminogen-1, TIMP-1 and -2 tissue inhibitors of metalloproteinases (MMPs), and nm23-H1, an anti-metastatic protein [96] (Figure 7).

Moreover, alpha-carotene was able to induce a reduction in Rho and Rac1 protein expression through the modulation of focal adhesion kinase-mediated phosphorylation of the mitogen-activated protein kinase family [96].

The anti-metastatic effects of β-carotene have also been investigated on the highly malignant SK-N-BE(2)C neuroblastoma cell line in vitro and in vivo. Treatment of SK-N-BE(2)C cells with β-carotene reduced the migratory and invasive capabilities of these cells. It was demonstrated that the β-carotene treatment of cells, under both normoxic and hypoxic conditions, inhibited the enzymatic activity and expression of matrix metalloproteinase (MMP)-2. Interestingly, β-carotene, in in vivo experiments, significantly reduces the incidence of liver metastases and the mean volume of neuroblastoma [97]. These data suggested, for the first time, that β-carotene could be used as an effective chemopreventive agent, capable of regulating neuroblastoma invasion and metastasis via HIF-1α [97].

Moreover, it has been observed that 6-methoxymellein, a dihydroisocoumarin belonging to the phenolic compounds found in carrots, not only inhibited cell proliferation but also acted as an inhibitor of cell migration in breast cancer cells. It was also able to reduce the expression of c-Myc, Sox-2, and Oct4, which are stemness-associated proteins, in breast cancer cells and to decrease the proportion of CD44+/CD24− breast cancer stem cells in the tumor population, thus inhibiting tumor progression [94]. 

Interestingly, it has been demonstrated that falcarindiol inhibits the growth of neural stem cells by suppressing the Notch pathway, suggesting that this compound may inhibit the growth of cancer stem cells [98]. The Notch pathway is critical to the tumorigenicity of cancer stem cells, which are the driving force of tumor progression [99].

Furthermore, it has been observed that carrot pectic polysaccharide interfered with galectin-3- and galectin-3 binding protein (G3BP)-mediated metastasis. It has been demonstrated that dietary carrot pectic polysaccharide (CRPP) was able to reduce the expression of galectin-3 and G3BP by about 80%. Moreover, this reduction in galectin3/G3BP expression was followed by the infiltration of macrophages into the tumor site, suggesting that carrot pectic polysaccharide inhibits metastasis through immunomodulation [100].

Additionally, pentane/diethyl ether (50:50) extracts from the carrot have been demonstrated to show a pronounced reduction in cancer cell motility as well as in cancer cell invasion on four different cancer types’ cell lines (lung, skin, breast, and glioblastoma cancer cells) [101]. 

### 3.3. Carrot Compounds and Their Role in Multidrug Resistance, MDR

It has been observed that compounds from carrots are able to modulate the human P-glycoprotein (P-gp) efflux function, the major ATP-binding cassette (ABC) efflux transporter [102], as shown in Figure 8.

Several efforts have been made in the development of various ABC efflux transporter inhibitors. However, none of them is able to provide better clinical outcomes for cancer patients, mainly due to systemic toxicities, as well as significant drug–drug interactions. 

Interestingly, compounds from carrots have been reported that may help in overcoming the multidrug resistance (MDR) issue, one of the major obstacles to successful cancer therapy, by stimulating basal and drug-induced P-gp ATPase activity. For instance, the addition of β-carotene to chemotherapeutic agents significantly enhanced the chemotherapeutic cytotoxicity in cancer cells in vitro, supporting the role of β-carotene as a chemo-sensitizer and suggesting its potential use in adjuvant therapy for cancer treatment [103]. 

Moreover, it has been demonstrated that falcarinol and falcarindiol polyacetylenes are able to inhibit ABCG2, the efflux protein also known as BCRP (breast cancer resistance protein) involved in breast cancer chemotherapy resistance, and mitochondrial aldehyde dehydrogenase (ALDH2) by covalently binding to these proteins [104]. In addition, it has been reported that the ALDH2 reduced ALDH2 activity may be associated with oxidative and endoplasmic reticulum (ER) stress, leading to cell cycle arrest and apoptosis [105,106]. Moreover, the FaOH and FaDOH ability to activate the Keap1-Nrf2-signaling pathway [81,107,108], due to their electrophilic nature, may explain, to some extent, the chemopreventive effects of these polyacetylenes. In fact, the Keap1-Nrf2 pathway is involved in the regulation of the expression and formation of various antioxidant, anti-inflammatory, and cytoprotective phase 2 enzymes.

### 3.4. Carrot Compounds and Their Role as Lipid Metabolism Modulators in Cancer

The role of various carrot compounds as lipid metabolism modulators in cancer cells has also been reported. Tumors show increased levels of cholesterol compared to normal tissue. In fact, in order to proliferate fast, cancer cells need high levels of cholesterol for membrane biogenesis and other functions. The excess cholesterol is accumulated and stored in cancer cells in the form of cholesterol esters within lipid droplets (LDs), which serve as a readily available reservoir for neutral lipids [109,110]. 

Furthermore, intracellular cholesterol accumulation has been reported to inhibit tumor cell death [111], suggesting that cholesterol reduction in tumor cells could be used as a potential anticancer strategy [111,112]. ABCA1, the ATP-binding cholesterol A1 transporter/exporter in human cancer cells, has been shown to reduce the amount of mitochondrial cholesterol, resulting in the release of molecules that promote cell death [113].

Among carrot metabolites, FaDOH has been reported to increase the lipid content in human mesenchymal stem cells (hMSCs) as well as the number of LDs. This effect appears to be associated with the increased expression of peroxisome proliferator-activated receptor gamma (PPARγ2), which in turn may induce the increased expression of ABCA1, the A1 transporter/cholesterol exporter, leading to endoplasmic reticulum stress and colon [84] and breast [59] cancer cell death.

## 4. Discussion 

Carrots are popular vegetables and cultivated all over the world. Several studies have been reported on their cultivation, breeding, nutrition content, tissue culture, and molecular research [4]. They are a good source of nutritious substances, such as vitamins, minerals, and dietary fiber. Additionally, carrots contain abundant biologically active substances, and several of them are important for human health [4]. 

Here, we describe the functional properties of the bioactive compounds extracted from carrots, with a particular focus on their anti-proliferation, pro-apoptotic, and anti-metastatic properties in different types of cancer cells, as summarized in Table 2. 

These properties make these compounds very attractive as potential candidates for cancer treatment, although contrasting data have been reported for some of them. Therefore, a lot still needs to be investigated to move forward in this direction. 

First, more investigations into the major bioactive compounds in different tissues and carrot genotypes, as well as in their tissue-dependent accumulation, need to be performed in order to design the optimal strategies for the processing and extraction of targeted bioactive compounds from the different tissues of given genotypes of carrot [41]. The extraction procedure also needs to be tailored to enrich the extract’s content in bioactive substances [86], and encapsulation techniques for potential compounds need to be developed in order to preserve the anticancer properties of these compounds, critical for their use in cancer treatment [114].

Two-dimensional cell systems have been helpful in highlighting the functional properties of bioactive substances from carrot extracts. However, these systems do not provide data on the way that different compounds are adsorbed by cancer cells in vivo and thus on the actual concentrations needed to make them work in vivo. In the case of β-carotene, its uptake by prostate cancer cells was observed to be strongly cell-line-dependent and androgen-sensitive-dependent [89]. Further, this study does not describe how the bioactive compounds’ uptake is affected by the tumor microenvironment (TME) if it is TME-dependent. 

Therefore, more studies are needed on the working mechanisms used by these compounds to act against tumor cells. For instance, a heated debate is ongoing on whether antioxidants (i.e., β-carotene) can be recommended to cancer patients who receive free-radical-producing radiotherapy or chemotherapy [115]. Conflicting data are reported in the literature on the role of antioxidants in cancer. In fact, while it has been reported that antioxidants are able to decrease the side effects of radiotherapy and chemotherapy with their ability to repair tissue damage caused by free radicals generated during anticancer therapy, other studies have reported that antioxidants can increase the risk of cancer. For instance, antioxidants have been reported to increase melanoma metastasis in mice [116] by protecting tumor cells from oxidative stress [117].

Of interest, data from preclinical trials where acetylenic oxylipins were tested demonstrated their anticancer properties and did not show any significant toxic side effects, suggesting their potential use in the prevention and treatment of cancers and as the main compounds from carrots for the development of anticancer drugs. However, only a few cytotoxic polyacetylenes have been tested in preclinical trials, and, so far, none of them has been tested in clinical trials, although there is evidence for a likely cancer-preventive effect of dietary polyacetylenes from meta-analysis and cohort studies [67]. 

Although highly bioavailable, these compounds are unstable, being sensitive to heat, light, and oxidation [50], and this may be one of the explanations for the relatively few preclinical trials performed on these compounds. 

Recently, more bioactive compounds belonging to the PUFA class have been isolated from carrots and showed a selective anti-proliferation effect on cancer cells, although further studies are needed to understand how these compounds act specifically in tumor cells [86].

Moreover, structure-based virtual screening, molecular docking studies, and advanced molecular dynamics simulations may be useful tools to provide further insights into the anticarcinogenic mechanisms of bioactive compounds from *Daucus carota* L., as demonstrated for anilinonaphthalene and the polyphenols curcumin and xanthohumol compounds [95,118].

Therefore, although all of these bioactive substances in carrots have good research value and prospects for medicinal use, more work still needs to be performed in the future to gain more evidence on their anticancer properties, including their pharmacokinetics and mechanisms of action, as well as their potential targets in cells.

## Figures and Tables

**Figure 1 molecules-28-07161-f001:**
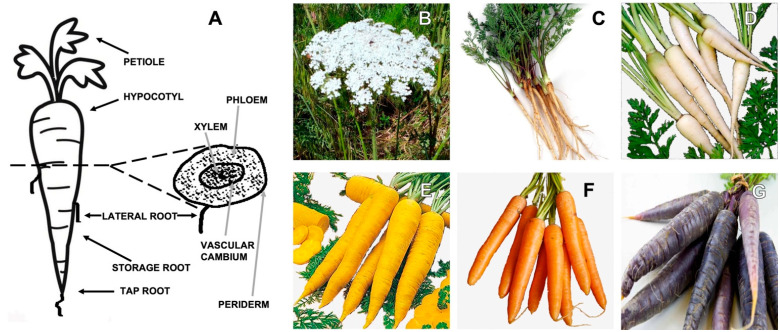
(**A**) Schematic representation of *Daucus Carota* L. Its different parts are reported: petiol, hypocotyl, and root and its cross-section, where the phloem and xylem are represented. (**B**) Carrot white flowers; (**C**) *D. carota* subsp. *carota* (wild carrot); (**D**) *D. carota* subsp. *Sativus* (domestic white carrot); (**E**) *D. carota* subsp. *Sativus* (domestic yellow carrot); (**F**) *D. carota* subsp. *Sativus* (domestic orange carrot); (**G**) *D. carota* subsp. *Sativus* (domestic violet carrot).

**Figure 2 molecules-28-07161-f002:**
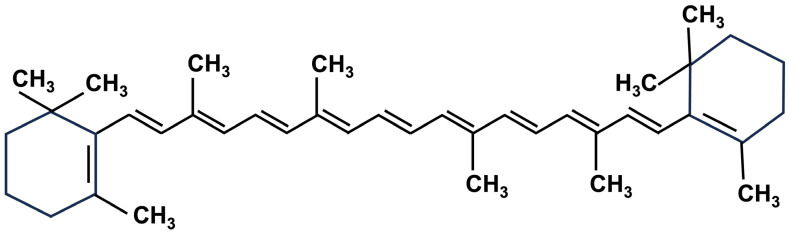
Structure of β-carotene.

**Figure 3 molecules-28-07161-f003:**
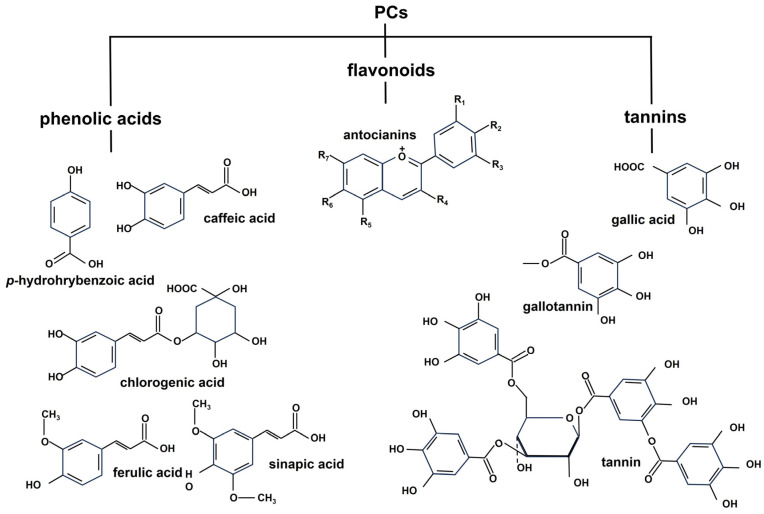
Structures of PCs. Here, the main PCs, flavonoids, tannins, and phenolic acids, are reported.

**Figure 4 molecules-28-07161-f004:**
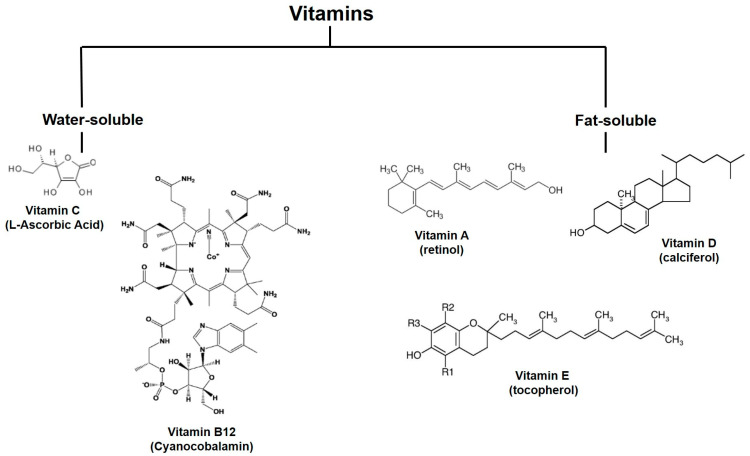
Structures of vitamins. Here, water- and fat-soluble vitamins are reported.

**Figure 5 molecules-28-07161-f005:**
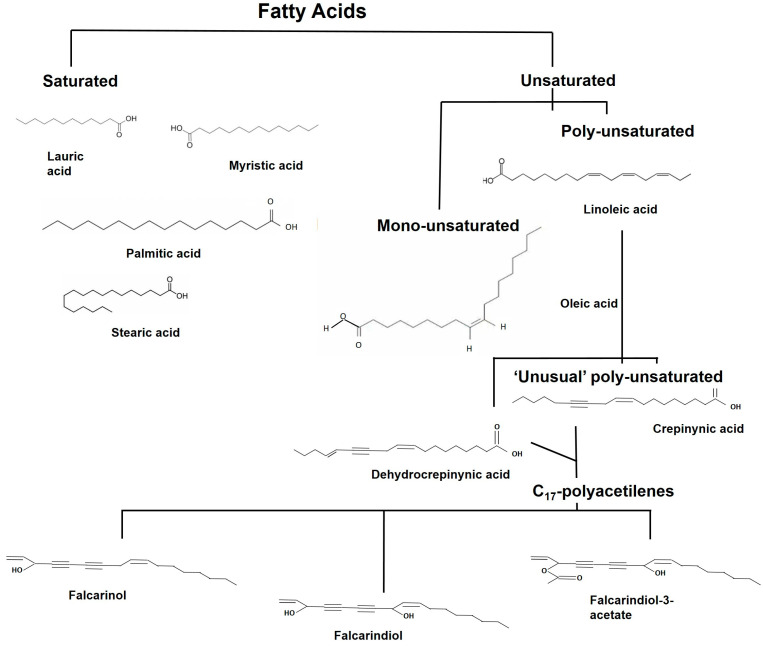
Structures of fatty acids and their derivatives. Here, the main fatty acids (FAs) present in carrots are reported: saturated FAs, unsaturated FAs, “unusual” unsaturated fatty acid precursors of the acetylenic oxylipins, and the most common falcarinol-type oxylipins.

**Figure 6 molecules-28-07161-f006:**
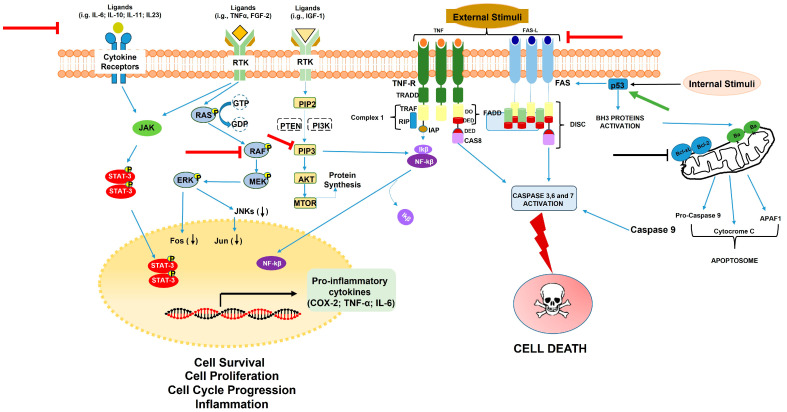
Oncogenic signaling and its modulation by carrot compounds. In this simplified cartoon, the MAPK/PI3K pathways involved in cell survival, proliferation, and cell cycle progression, as well as the apoptosis (intrinsic and extrinsic routes) and inflammation pathways, are illustrated. The influence of carrot compounds is represented: a stimulatory effect is indicated by a green arrow, while suppression is represented by a red vertical line with a small horizontal line. The downward black arrows between brackets indicate a downregulation effect induced by these compounds.

**Figure 7 molecules-28-07161-f007:**
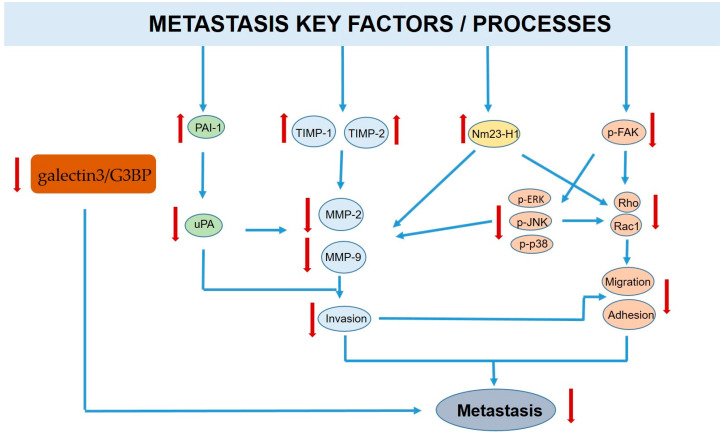
Carrot compounds and their metastasis regulation. Main key factors/processes leading to metastasis are reported. The upward and downward red arrows indicate the effect of upregulation and downregulation mediated by the carrot compounds, respectively, on the factors indicated.

**Figure 8 molecules-28-07161-f008:**
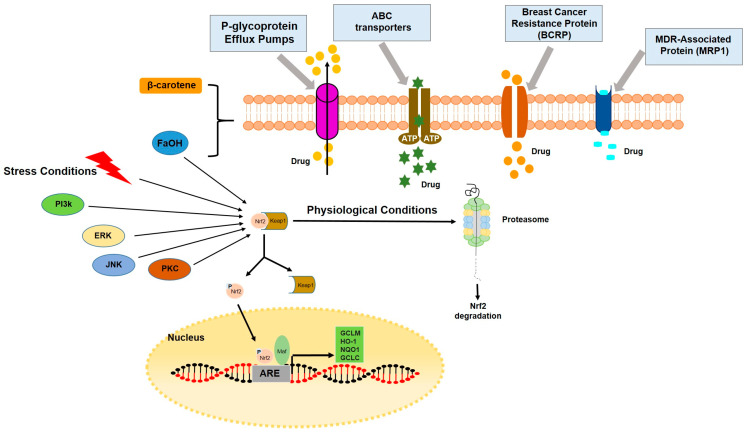
Carrot compounds and multidrug resistance, MDR. The more studied ATP-binding cassette (ABC) efflux transporters belonging to the superfamily of ABC proteins are reported: P-glycoprotein (P-gp), ABCG2/BCRP, and MRP1. Moreover, the Keap1-Nrf2 pathway is also illustrated.

**Table 1 molecules-28-07161-t001:** Chemical constituents of carrot.

Chemical Constituents	Observed Amount	References
Moisture	86–89 mg/100 g FW	[7]
Ca	34–80 mg/100 g FW	[7]
P	25–53 mg/100 g FW	[7]
K	240 mg/100 g FW	[7]
Mg	9 mg/100 g FW	[7]
Mn	0.2–0.8 mg/kg FW	[8]
Fe	0.4–2.2 mg/100 g	[7]
Na	40 mg/100 g	[7]
Total sugars	2.73–11.24 g/100 g FW	[8]
Total organic acids	1.07–2.79 g/100 g FW	[8]
Vitamin C (ascorbic acid)	1.0–5.3 mg/100 g * FW	[8]
Total phenolics	7.3–224 mg/100 g FW	[8]
Tetraterpenoids (carotenoids, chlorophylls)	0.2–4.1 mg/100 g FW	[8]
Falcarinol ^§^	16–84 mg/kg FW	[9]
(C17-polyacetilens) falcarindol ^§^	8–27 mg/kg FW	[9]
Falcarindol-3acetate ^§^	8–40 mg/kg FW	[9]

FW = fresh weight. * Depending on the storage time and conditions (after 30 days, the value could reduce to 50% of the initial value). ^§^ From cultivated orange carrot (*D. carota* ssp. *sativus*).

**Table 2 molecules-28-07161-t002:** Carrot compounds and their roles in cancer.

Compounds	Biological Effects in Cancer	References
α-Carotene (AC)	Anti-proliferative, anti-metastatic	[88,96]
β-Carotene (BC)	Anti-proliferative, anti-inflammatory, anti-apoptotic, anti-metastatic, ABC efflux transporter modulator	[89,90,91,92,97]
Antocianins	Cytotoxic	[93]
6-Methoximellein	Anti-inflammatory, anti-metastatic	[94]
Falcarinol (FaOH)	Cytotoxic, ABC efflux transporter modulator, anti-inflammatory	[55,57,58,65,66,77,81]
Falcarindiol (FaDOH)	Anti-inflammatory, cytotoxic, ABC efflux transporter modulator, lipid modulator,	[59,65,66,84,98,99,108]
Glycerophosphocoline derivatives (GPC)	Anti-proliferative	[86]
Monogalactosyl-monoacylglycerol (MGMG)	Anti-proliferative	[86]
Carrot pectic polysaccharide (CRPP)	Anti-metastatic	[100]

## Data Availability

Not applicable.

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
