# Peer review of "Chemical Composition, Functional and Anticancer Properties of Carrot"

_molecules, 2023, doi:10.3390/molecules28207161_

Round 1

Reviewer 1 Report

This review focus on compounds from carrots with anticancer properties. Of the compounds with potential anticancer properties that is discussed in detail are carotenoids, polyphenols incl. flavonoids and phenolic acids, vitamins, and fatty acids. However, the acetylenic oxylipins falcarinol and falcarindiol and their derivatives that have shown interesting anti-inflammatory and anticancer activity both in vitro and in vivo in numerous investigations are only mentioned on page 12, line 491-499 in the review. These highly interesting bioactive compounds are fatty acid derivatives and are more interesting than common fatty acids discussed in section 2.4. In addition, it is mentioned in the review that falcarinol and falcarindiol are produced by carrots in response to fungal diseases acting as natural pesticides on page 10, line 393-394. This can be misunderstood because the compounds are present in carrots before fungal infection but may increase in response to fungal diseases. Therefore, acetylenic oxylipins also need to be included in Table 1 as important chemical constituents in carrots and I clearly recommend based on the title of the review that the anticancer activity of acetylenic oxylipins such as falcarinol and falcarindiol are discussed in much more detail in the review.

Carotenoids are important nutraceuticals but in relation to the anticancer properties of carrots it is a controversial topic because there are also several investigations that have shown that carotenoids from carrots in fact increase the risk of cancer in high-risk groups. In addition, in Figure 1, white carrots are shown, and they do not contain carotenoids or at least very minute amounts. How does this fit into the long description of carotenoids being very important compounds in carrots contribution to the anticancer properties of carrots? My suggestion is that the section about the anticancer activity of carotenoids is written with a more critical approach in relation to the anticancer effect of carrots.

Polyphenols from carrots have not shown to be important contributors in relation to the anti-inflammatory and/or anticancer effect of carrots. Polyphenols contribute to the direct antioxidant activity in vitro but this is not the same that they contribute to the antioxidant in vivo. This is highly important. For example, falcarnol and falcarindiol are not antioxidants in vitro but are able to stimulate Keap1-Nrf2-ARE pathway in cells and thus are able to stimulate the formation of cytoprotective enzymes (phase 2 enzymes) and are considered as compounds with antioxidant activity in vivo as well as immunostimulators etc. Consequently, the section about polyphenols also needs some modification and a more critical approach toward the role of these compounds in the anticancer properties of carrots.

In conclusion, this manuscript needs to be rewritten with a more critical approach towards the compounds discussed in the review in relation to their possible contribution to the anticancer properties of carrots.   Furthermore, the contribution of acetylenic oxylipins to the anticancer properties of carrots also need to be discussed in more detail in this review.

Author Response

Dear Referee,

We appreciated your valuable suggestions.

In the text, we reported the multiple roles in cancer of acetylenic oxylipins, specifically in the section: 3.1 Carrot compounds and their role in cell survival, proliferation and apoptosis, page11, line 491-502; in section 3.3. Carrot compounds and their role in the MultiDrug Resistance, MDR page14, line 596-606 and in the section 3.4. Carrot compounds and their role as lipid metabolism modulators in cancer page 15, line 620-625. In addition, we have also cited a very exhaustive review, that has been published on the topic, (see ref. 60 Christensen, L.P. Bioactive C17 and C18 Acetylenic Oxylipins from Terrestrial Plants as Potential Lead Compounds for Anticancer Drug Development. Molecules 2020, 25, 2568), where the anti-cancer role of these compounds isolated from different plants, enclosed Carrot, has been extensively illustrated.

Moreover, we added more details about these interesting compounds (the acetylenic oxylipins falcarinol and falcarindiol and their derivatives). We have added more information about their content in carrot in section 1.1 Daucus carota L. chemical composition (page 4, line 171-176). We also listed them in Table 1 on page 3.

We insert more detail on their structure in section 2.4. Fatty acids on page 10, line 419-422 and line 426-428 on page 11, as well as in Figure 5, that has been modified.

In addition, to avoid misunderstanding about the role of these compounds as natural pesticides, we have changed the sentence on page 12, line 492-494:

 ‘They are also known as the polyacetylenic oxylipins and they are produced by carrot in response to fungal diseases, acting as natural pesticides…..’

With:

‘They are also known as the polyacetylenic oxylipins and it has been observed that their expression is increased by carrot in response to fungal diseases, acting as natural pesticides …..’

In reference to the fact that phenolic compounds from carrots have not shown to be important contributors in relation to the anti-inflammatory and/or anticancer effect of carrots, it is not completely right.

Yes, they contribute to the direct antioxidant activity in vitro which is not the same to the one in vivo (flavonoids, anthocyanins) but they also show anti-inflammatory and/or anticancer effects. Thus, we reported more in detail that 6-Methoxymellein, a phenolic compound, exhibiting anti-inflammatory and anticancer properties (see section 3.1 Carrot compounds and their role in cell survival, proliferation and apoptosis, page 12, line 488-490), and previously cited only in section 3.2. Carrot compounds and their anti-metastatic role, page 13, line 561-566.

               Finally, white carrots illustrated in Figure 1, represent just an example of Carrot varieties described in the introduction section.

Reviewer 2 Report

In the submitted manuscript Mandrich et. al. reviewed the chemical composition and biological activities of edible and wild Daucus carota L. subspecies. The main focus of the manuscript was on the main bioactive compounds in carrots highlighting their anti-cancer properties.

The manuscript is well-written and relevant to the field. In my opinion, the submitted manuscript could be accepted for publication in Molecules. However, there are a few comments that should be addressed before its publication:

Section 2. Daucus carota L. bioactive components with health-promoting function or phytochemicals

A new table summarizing the observed biological activities of bioactive compounds from Daucus carota L. would be beneficial to the readers.

The authors should also discuss the potential of bioactive compounds in the development of novel anti-cancer therapeutic strategies in more detail. The review of state-of-the-art extraction and encapsulation techniques would be of added value to the manuscript.

Reference:

1. Furlan, V.; Bren, U. Helichrysum italicum: From Extraction, Distillation, and Encapsulation Techniques to Beneficial Health Effects. Foods 2023, 12, 802.

Section Discussion

Line 539: A brief review of mechanistic in silico studies should be provided. Recently new inverse molecular docking protocol as well as advanced molecular dynamics simulations were applied to identify potential protein targets of polyphenols curcumin and xanthohumol, which could also provide further insights into anticarcinogenic mechanisms of bioactive compounds from Daucus carota L.

Reference:

1. Pantiora, P.; Furlan, V.; Matiadis, D.; Mavroidi, B.; Perperopoulou, F.; Papageorgiou, A.C.; Sagnou, M.; Bren, U.; Pelecanou, M.; Labrou, N.E. Monocarbonyl Curcumin Analogues as Potent Inhibitors against Human Glutathione Transferase P1-1. Antioxidants 2023, 12, 63. https://doi.org/10.3390/antiox12010063

Author Response

Thank you for your valuable suggestions.

A new table (Table 2) summarizing the role of bioactive compounds from Daucus carota L. In cancer, has been added in the Discussion, page 15-16, as suggested.

We also discussed the need to develop not only tailored extraction procedures to enrich potential compounds but also develop efficacious encapsulation techniques to preserve the anti-cancer activity of these compounds, in discussion section, page 16, line 648-650, and we added the suggested reference (Furlan, V.; Bren, U. Helichrysum italicum: From Extraction, Distillation, and Encapsulation Techniques to Beneficial Health Effects. Foods 2023, 12, 802.)

Moreover, as suggested, we added in the discussion section, page 17, line 673-678 the need of structure-based virtual screening molecular docking studies and advanced molecular dynamics simulations as useful tools to provide further insights into anticarcinogenic mechanisms of bioactive compounds from Daucus carota L. and we also insert the suggested reference: Pantiora, P.; Furlan, V.; Matiadis, D.; Mavroidi, B.; Perperopoulou, F.; Papageorgiou, A.C.; Sagnou, M.; Bren, U.; Pelecanou, M.; Labrou, N.E. Monocarbonyl Curcumin Analogues as Potent Inhibitors against Human Glutathione Transferase P1-1. Antioxidants 2023, 12, 63.

Round 2

Reviewer 1 Report

Even though the revised manuscript has been improved, the authors have only made changes corresponding to minor revisions, although I asked for a major revision of the manuscript. Some of the most interesting compounds in relation to the anticancer effect of carrots are the acetylenic oxylipins falcarinol and falcarindiol, which have shown both cytotoxic and anti-inflammatory activities in vitro in numerous investigations as well as anti-inflammatory and anticancer effects in vivo. Still the bioactivity of these compounds is only mentioned/discussed very briefly in this review. Why use so much space on for example fatty acids when one of the major contributors to the anticancer effects of carrots have been shown to be due to the fatty acid derivatives falcarinol and falcarindiol and related acetylenic oxylipins. Consequently, the review still needs some revision on this point, i.e., revision of section 2.4 and 3. In addition, I suggest changing the headline of section 2.4 from “Fatty acids” to “Fatty acid and their derivatives”.  

The bioactivities of the different constituents in carrots discussed in section 3 are mainly in vitro studies with a few exceptions even though there exist in vivo studies for many of the constituents discussed in the review. For example, the anticancer effects of the acetylenic oxylipins have been described in rodents so why not discuss these results. Furthermore, there exist a clear connection between the development of cancer and inflammation and thus the anticancer effect of some of the constituents are also closely related to their anti-inflammatory activity. This is not clear from this review. Furthermore, there is a clear difference between the preventive effects of carrot constituents against cancer and the treatment of cancer. None of the constituents in carrots can be used to treat cancer but they may contribute to the cancer preventive effects of carrots. Still, the authors indicate for example on line 532-536 and 567-571 that beta-carotene can be used as a chemotherapeutic agent, and such statement clearly need to be substantiated by more data. In fact, there are in vivo studies that indicate that beta-carotene can increase cancer incidence in high-risk patients and therefore one should be careful to suggest that beta-carotene can be used in cancer treatment.

In conclusion, this manuscript still needs to be rewritten (comprehensive  revision), and especially section 3 and 4 with a more critical approach towards the bioactive compounds discussed in the review in relation to their possible contribution to the anticancer properties of carrots. 

Author Response

Dear Reviewer,

We appreciated your constructive criticisms. We rewrote the section 3 and 4 of the manuscript, addressing your concerns.

We have illustrated more in detail the anticancer effect of the acetylenic oxylipins, describing cytotoxic and anti-inflammatory activities observed in in vitro and in vivo studies and discussing their potentiality as anti-cancer drugs (section 3, lines from 462 to 576; 688-691; 735-745; 759-764 and in section 4, lines 809-822).

We changed the headline of section 2.4 from “Fatty acids” to “Fatty acids and their derivatives”.  

Furthermore, we clarified the existence of a clear connection between the development of cancer and inflammation and thus the anticancer effect of some constituents related to their anti-inflammatory activity.

We also included the inflammatory pathway, in Figure 6.

In addition, thanks for your point: ‘None of the constituents in carrots can be used to treat cancer but they may contribute to the cancer preventive effects of carrots. Still, the authors indicate for example on line 532-536 and 567-571 that beta-carotene can be used as a chemotherapeutic agent, and such statement clearly need to be substantiated by more data.’  We are agreed and we changed the chemotherapeutic word with ‘chemopreventive’ in the manuscript.

 Best Regards, 

 Emilia Caputo, PhD